# Learning Dual Text Embeddings by Synthesising Images Conditioned on Text

## Abstract

Text-to-Image (T2I) synthesis is a challenging task that requires modelling complex interactions between two modalities (*i.e.*, text and image). A common framework adopted in recent state-of-the-art approaches to achieving such multi-modal interactions is to bootstrap the learning process with pre-trained image-aligned text embeddings. These text embeddings are typically learned by training an independent network with a contrastive loss between text and image features. Such a scheme comes with the downside that these embeddings are learned to capture distinctive features and trained only to differentiate between instances. These learned text embeddings are unaware of the different perspectives of generation to capture intricate, complex variations of image generation and discrimination process to capture distinctive features, which may hinder their usage in generative modelling.

To alleviate this downside, this paper explores a new direction to learn text embeddings in an end-to-end manner from text-to-image synthesis task that considers the different perspectives of generation and discrimination process. Specifically, a novel text-embedding learning scheme called "Dual Text Embedding" (DTE) is presented, in which one part of the embeddings is optimised to enhance the photo-realism of the generated images, and the other part seeks to capture text-to-image alignment. Through a comprehensive set of experiments on three text-to-image benchmark datasets (Oxford-102, Caltech-UCSD, and MS-COCO), models with dual text embeddings perform favourably in comparison with embeddings trained only to learn distinctive features.

## 1 Introduction

Visualising images for any textual statement is elemental to human understanding of the world. Intelligent systems' ability to generate images from the text for human understanding has a wide range of applications such as information sharing, computer-aided design, text-to-image search and photo editing. Image synthesis from text is a challenging task due to complex interaction and ambiguous association of text modality with the image modality. For instance, multiple textual descriptions can describe the same image and vice versa. Further, finer details of the images may not always be well captured in textual descriptions. In this domain, Generative Adversarial Networks (GANs) (Goodfellow et al., 2014) were the go-to method for generating realistic images. Conditional GANs (Mirza & Osindero, 2014; Odena et al., 2017; Miyato & Koyama, 2018) allow us to generate real images semantically coherent with the text (Reed et al., 2016b; Dash et al., 2017; Reed et al., 2016a) by conditioning the generation process on global sentence embeddings.

Though GANs have been shown to generate meaningful images, directly generating high-resolution images from text leads to sub-par visual results and training instability due to the multi-modal nature (*i.e.,* image *vs.* text) of the task. One set of methods attempts to solve this issue by *advancing the visual generation* part of the model. For instance, StackGAN (Zhang et al., 2017b) employs a *hierarchical stage-wise training* of GANs from a low-resolution to a high resolution and conditions the generator at every stage by images generated from the previous stage generator.

Another set of methods (Xu et al., 2018; Zhu et al., 2019; Liang et al., 2020) addresses high-quality image generation from a text by improving the compatibility between text and image modalities. It is achieved

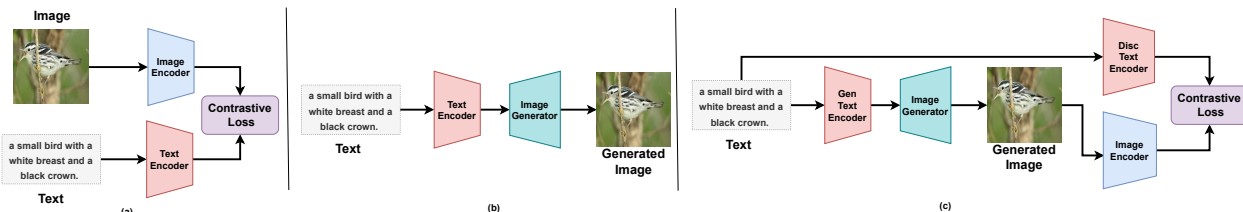

Figure 1: (a) Text and image are projected into a shared embedding space to enhance mutual information capturing discriminative features (Discriminative Embeddings), (b) Word embeddings are trained by generating images capturing semantic details (Generative Embeddings) and (c) Dual Text Embeddings approach combining both generative and discriminative embeddings.

by semantically aligning the visual features in image sub-regions with the pre-trained word embeddings through attention. Among these methods, text embeddings trained by projecting text and image features into common embedding space and increasing the mutual information between the text-image pairs are shown to be crucial for high-quality image generation. Recent state-of-the-art methods (Xu et al., 2018; Liang et al., 2020; Tao et al., 2022; Liao et al., 2022; Ramesh et al., 2021; 2022; Gafni et al., 2022; Rombach et al., 2021) are predominantly trained using pre-trained vision-language models (Radford et al., 2021; Xu et al., 2018). These vision-language models project images and text into a shared embedding space, where matching pairs exhibit high similarity and dissimilar pairs show low similarity. This training relies on contrastive learning to distinguish between pairs by learning discriminative features as shown in Figure 1.a, resulting in superior text-image alignment. Such models are commonly used for encoding captions in Text-to-Image synthesis. This scheme of learning embeddings comes with a limitation that these learned text embeddings are unaware of the different perspectives of generation and discrimination process, *i.e.*, the generative process necessitates learning complex appearance variations needed for creating realistic images. Integrating generative and discriminator perspectives leads to a superior text encoding scheme, enhancing Text-to-Image synthesis.

In this work, we explore a new direction to capture both such perspectives in an unified end-to-end training as shown in Figure 1.c for improving compatibility between image and text modalities. In this regard, a novel *Dual Text Embedding GAN (DTE-GAN)* setup is proposed to learn text embeddings in an end-to-end manner that takes into account the different perspectives of the generation and discrimination process. Specifically, DTE contains two parts of embedding for each word: 1) *Generator-side embedding* to capture image generation specific characteristics and is specifically optimised to improve the quality of the generated images, and 2) *Discriminator-side embedding* seeks to increase text-to-image alignment and is learnt from the multi-modal contrastive loss between the image and text features. Owing to its simpler design, a single-stage GAN (Tao et al., 2022) is used for Text-to-Image synthesis. Specifically, the DTE-GAN uses a generator $G$ to synthesise images from the given text and a discriminator $D$ to give feedback on whether the generated image is real or fake and align the text and image pairs semantically through cross-modal contrastive loss (Zhang et al., 2021). Besides generating images consistent with the text, the model learns the dual-text embeddings (DTE) to capture the different perspectives of the generation and discrimination process. Naive approaches of independent or fully shared text embeddings between the generator and discriminator lead to sub-par results. We hypothesise that one of the prominent reasons for such sub-par results is the noisy image generation during early GAN training.

To evaluate the effectiveness of the proposed approach, experiments are conducted on three datasets namely, 1) Oxford-102 (Nilsback & Zisserman, 2008), 2) Caltech-UCSD Birds 200 (CUB) (Welinder et al., 2010) and 3) MS-COCO (Lin et al., 2014b). Three metrics are used for quantitatively evaluating the generated images: *Inception Score* (Salimans et al., 2016) and *Fréchet Inception Distance (FID)* (Heusel et al., 2017) for quality of the images and *R-precision* (Xu et al., 2018) to measure the semantic consistency between the generated image and the text. Our model decreases the FID score from 14.06 to 13.67 and 40.31 to 30.07 on the CUB and Oxford-102 datasets respectively. For the COCO dataset, our model decreases the FID

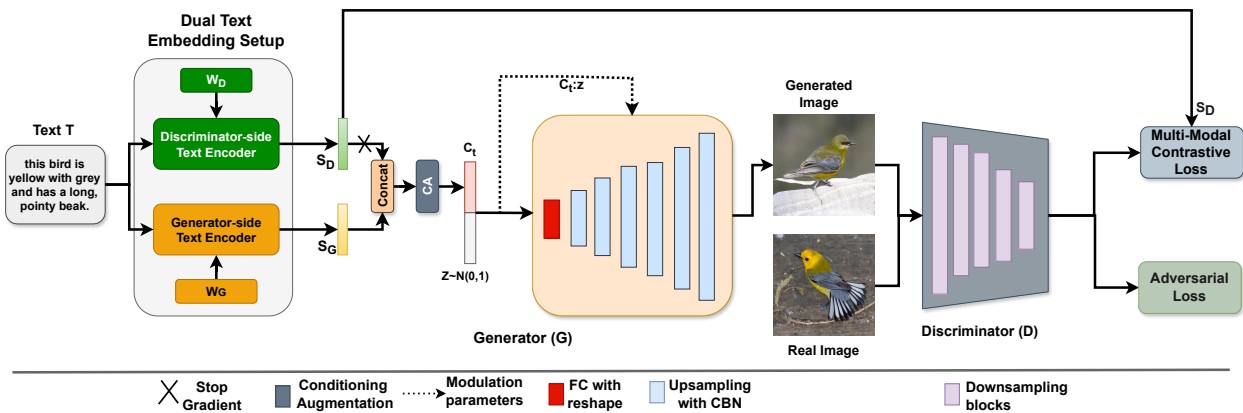

Figure 2: Overview of DTE-GAN architecture. DTE-GAN consists of three core components: i) a single-stage generator $G$ (Section 3.2), ii) a discriminator $D$ (Section 3.3), and iii) a dual text embedding setup (Section 3.4). In the Figure, $W_G$ = generator-side word embeddings, $S_G$ = generator-side sentence embedding, $W_D$ = discriminator-side word embeddings, $S_D$ = discriminator-side sentence embedding. The model is optimised using two objective functions: 1) adversarial loss, and 2) multi-modal contrastive loss.

from 35.49 to 25.17 in comparison to that of AttnGAN (Xu et al., 2018) which uses embeddings trained to capture distinctive features. Contribution of this paper are summarised as follows:

- A novel framework to learn dual text embeddings to encompass the different perspectives of the generation and discrimination process is proposed.

- DTE setup is incorporated into a single-stage GAN to learn text embeddings and generate images simultaneously in an end-to-end framework.

- Through a comprehensive set of experiments and ablation studies, performance improvements of DTE approach are illustrated comparing with those using embedding trained to capture distinctive features only.

## 2    Related Work

In this section, some of the relevant works in the literature relating to this paper are discussed briefly.

**Generative Adversarial Networks:** In past few years, GANs (Goodfellow et al., 2014) had been the go-to method for generating images and class-specific images (Mirza & Osindero, 2014; Odena et al., 2017; Miyato & Koyama, 2018) on small datasets such as MNIST (LeCun & Cortes, 2010) and CIFAR (Krizhevsky et al.). However, GAN training is highly unstable when used to generate images on large datasets such as ImageNet (Deng et al., 2009). Researchers have explored to fix this training instability by re-framing GAN loss and regularisation (Arjovsky et al., 2017; Gulrajani et al., 2017; Mao et al., 2017; Miyato et al., 2018; Brock et al., 2017) to generate high-resolution images on large datasets (Karras et al., 2018; Brock et al., 2019).

**Text-to-Image synthesis:** GANs conditioned on global sentence-level embeddings are known to generate meaningful images at low resolutions (Reed et al., 2016a; Dash et al., 2017; Reed et al., 2016b). StackGAN (Zhang et al., 2017b) generates high-resolution images in stage-wise approach, where the generator at each stage is conditioned by the image generated from the previous stage. Unlike StackGAN, HDGAN (Zhang et al., 2018) trains single generator and multiple discriminators for each resolution. AttnGAN (Xu et al., 2018) uses text embeddings for fine-tuning the image features and also introduces a multi-modal contrastive loss (DAMSM loss) to bridge the gap between generated images and words. DM-GAN (Zhu et al., 2019) refines words and image features using a memory module. MirrorGAN (Qiao et al., 2019b) generates caption

for generated images that improves the text *vs.* image semantic consistency. SD-GAN (Yin et al., 2019) introduces Siamese structure for generator that uses Conditional Batch Normalization (CBN) (Chen et al., 2019) to improve the text-image alignment. CPGAN (Liang et al., 2020) learns a memory attended text encoder by attending to salient features in images for each word and fine-grained discriminator (Li et al., 2019). DTGAN (Zhang & Schomaker, 2020) applies channel and spatial attention, conditioned on sentence vector to focus on important features for each textual representation. XMC-GAN (Zhang et al., 2021) maximises the mutual information between text and image using intra-modality and inter-modality contrastive losses. DF-GAN (Tao et al., 2022) uses deep affine transformed global sentence embedding to condition the generator and matching-aware discriminator. In this space of text-to-image semantic alignment-based methods, pre-trained embeddings are an inherent prerequisite. These embeddings are only trained by discriminative approach. Unlike these methods, DTE attempts to learn text embeddings that capture generative and discriminative properties.

**Generative embedding learning:** Some methods attempt to learn embeddings (text / visual) end-to-end as part of a generator. For instance, (van den Oord et al., 2017; Esser et al., 2020) learn discrete embeddings for visual representation and show substantial improvement in Text-to-Image synthesis performance (Ramesh et al., 2021; Ding et al., 2021). Further better and compact representation are learned to improve the quality of image generation (Razavi et al., 2019; Lee et al., 2022; Gafni et al., 2022). Unlike these works that consider only the generation process while learning embeddings, DTE explores to capture different perspectives of generation and discrimination process by learning dual text embeddings.

**Large Scale Text-to-Image Synthesis:** Denoising Diffusion Probabilistic models (Sohl-Dickstein et al., 2015) are currently achieved remarkable success in image generation (Ho et al., 2020; Nichol & Dhariwal, 2021; Dhariwal & Nichol, 2021) by reversing the *forward markovian process* with noise removal in multiple steps. Though diffusion-based models are able to generate images with complex and varied interactions for text and generate high-quality images (Ramesh et al., 2022; Saharia et al., 2022; Gu et al., 2022), these approaches require large-scale training and exploit pre-trained discriminative language models like CLIP (Radford et al., 2021). Further, CLIP-based language and image encoders are used as a bootstrapping approach for predicting conditional representations in large-scale GAN-based approaches (Tao et al., 2023; Kang et al., 2023; Zhou et al., 2021). Unlike CLIP, which is trained only to capture distinctive representations, we propose DTE to capture the different perspectives of generation and discrimination.

## 3 Methodology

In this section, an end-to-end framework called "*Dual Text Embedding GAN*" (DTE-GAN) is formulated to learn dual text embeddings and capture the different perspectives of the generation and discrimination process in the text-to-image synthesis task. In the following sub-sections, overall architecture is introduced followed by specific details on generator and discriminator architectures respectively; then the final sub-section focuses on learning dual text embeddings.

### 3.1 Model overview

DTE-GAN consists of three core components: 1) a novel dual text embedding setup, 2) a single-stage generator $G$, and 3) a discriminator $D$. The overall architecture is shown in Figure 2.

First, the given text $T$ is passed through the novel dual text embedding procedure to encode the text into two types of sentence embeddings, namely: 1) Generator-side sentence embedding $S_G$, and 2) Discriminator-side sentence embedding $S_D$. The dual text embedding setup consists of two separate Bi-LSTM (Schuster & Paliwal, 1997) text encoders (*Generator-side* and *Discriminator-side*) and their own independent word embeddings ($W_G$ & $W_D$). As the name suggests, the *Generator-side* word embeddings $W_G$ and its encoder are intended to be trained from the image-generation process (generator $G$) and its losses, while the *Discriminator-side* word embeddings $W_D$ and its encoder are optimised by the loss from the multi-modal contrastive branch of discriminator $D$. Such a decoupling between these two parts of embedding adds flexibility in capturing different natures of image generations and discrimination processes. Specifically, the image generation process warrants learning specific appearance details along with the intricacies involved

in creating an image, while the discrimination process strives to learn features to simply differentiate the instances. Further, the separation of these two embeddings allows $W_G$ to learn from noisy gradients of $G$ independently (as $G$'s gradients are initially noisy due to fake image - sentence pairs), while $W_D$ learns from stable gradients of $D$ (real image - sentence pairs).

During training, $W_G$, the generator-side text encoder, and Generator $G$ are trained from gradient signals of the generation process *i.e.,* Adversarial loss of fake images ($I_{fake}$) and multi-modal contrastive loss between the generated image $I_{fake}$ and the given text $T$. Next, $W_D$ and the discriminator-side text encoder are trained from the gradient signals of the multi-modal contrastive loss between the real image $I_{real}$ and the given text $T$. Further, discriminator $D$ is trained from Adversarial loss ($I_{fake}$, $I_{real}$) and multi-modal contrastive loss between the real image $I_{real}$ and the given text $T$.

## 3.2 Generator

As opposed to other methods that use stacks of GANs or hierarchical GAN, a single-stage generator is employed that can generate an image at any resolution owing to its simpler design and easier training procedure. The generator $G$ takes three inputs: i) a noise vector $z$ of dimension $d_z$ from a Standard Gaussian Distribution $\mathcal{N}(0, 1)$ with the Truncation Trick (Brock et al., 2019; Tao et al., 2022; Zhang & Schomaker, 2020), ii) the generator-side sentence embedding $S_G$ of dimension $d_{SG}$, and iii) the discriminator-side sentence embedding $S_D$ of dimension $d_{SD}$. Next, the two sentence embeddings ($S_G$, $S_D$) together are passed through the conditioning augmentation (Zhang et al., 2017b) to get the conditional vector $C_t$ which is concatenated with a noise vector $z$ sampled from a Standard Gaussian Distribution $\mathcal{N}(0, 1)$ to form the input vector $f_G$ and passed through a fully connected layer with reshape to create a low-resolution spatial feature map. It may be noted that, in this step, $S_D$ is detached from the gradient flow of $G$ to avoid getting gradients from the generation process (refer to ablation studies in Section 4.3.4).

Further, this low-resolution feature map is passed through a set of upsampling blocks (*UpBlock*) followed by a convolution layer that accepts the last high-resolution feature map and outputs the generated image $I_{fake}$ of dimension $3 \times h \times w$ ($h$ = height, $w$ = width). Each *UpBlock* is formulated as a residual layer consisting of a bi-linear upsampling step followed by two convolution blocks (convolutional layer + Conditional Batch Normalization (CBN)(Chen et al., 2019) + LeakyReLU (Maas et al., 2013)). To increase the stochastic capability of the model, scaled noise is added (similar to StyleGAN (Karras et al., 2019)) to input before passing to the convolutional layer. The modulation parameters in Conditional Batch Normalization (CBN) (Yin et al., 2019; Chen et al., 2019) $\gamma_c$, and $\beta_c$ are calculated from $f_G$ by means of a linear projection layer. The modulation parameters $\gamma_c$, and $\beta_c$ in CBN are calculated as follows:

$$\text{BN}(x \mid C_t, z) = (\gamma + \gamma_c) \cdot \frac{x - \mu(x)}{\sigma(x)} + (\beta + \beta_c) \tag{1}$$

$$f_G = \text{Concat}[C_t, z] \tag{2}$$

$$\gamma_c = FC_\gamma(f_G) \tag{3}$$

$$\beta_c = FC_\beta(f_G) \tag{4}$$

The generator is trained to minimise adversarial loss ($\mathcal{L}_{Adv}^G$), and multi-modal contrastive loss ($\mathcal{L}_{cont}^G$). $\mathcal{L}_{cont}^G$ is formulated as a loss between the features from the generated image and the discriminator-side sentence embeddings. Mathematically, the objective functions can be written as follows:

$$\mathcal{L}_{Adv}^G = \mathbb{E}_{\hat{x} \sim p_G}[-D(\hat{x})] \tag{5}$$

$$\mathcal{L}_{cont}^G \left( \hat{f}_{v_i}, S_{D_i} \right) = -\log \frac{\exp\left( Sim\left( \hat{f}_{v_i}, S_{D_i} \right) \right)}{\sum_{j=1}^N \exp\left( Sim\left( \hat{f}_{v_i}, S_{D_j} \right) \right)} \tag{6}$$

$$Sim(f_v, S_D) = \cos\left( f_v, S_D \right) / \tau \tag{7}$$

Here, $Sim(.,.)$ is a score function to calculate the similarity between sentence embeddings and image features, $\cos(u, v) = u^T v / \|u\| \|v\|$ is the Cosine Similarity between features and $\tau$ denotes the temperature hyper-parameter, and $\hat{f}_v$ represents visual features extracted by the discriminator for the generated image $I_{fake}$. We

use conditioning augmentation (Zhang et al., 2017b) to sample the sentence condition from an independent Gaussian Distribution $\mathcal{N}\left(\mu\left(s_t\right), \Sigma\left(s_t\right)\right)$. The regularisation term from conditioning augmentation ($\mathcal{L}_{CA}$) for combined sentence embeddings($s_t$) is:

$$\mathcal{L}_{CA} = D_{KL}\left(\mathcal{N}\left(\mu\left(s_t\right), \Sigma\left(s_t\right)\right) \| \mathcal{N}(0, I)\right) \tag{8}$$

Here $\mu(s_t)$ and $\Sigma(s_t)$ are mean and diagonal covaraince matrices that are computed as functions of the combined sentence embedding. The regualarisation term is KL Divergence between the Conditioning Gaussian and a Standard Gaussian Distribution. The final loss for generator is defined as:

$$\mathcal{L}_G = \mathcal{L}_{Adv}^G + \lambda_1 \mathcal{L}_{CA} + \lambda_2 \mathcal{L}_{\text{cont}}^G \tag{9}$$

### 3.3 Discriminator

The discriminator $D$ is designed to serve two purposes: (1) to be a critic to determine whether the image is real or fake, and (2) to be a feature encoder to extract image features for multi-modal contrastive loss. The given image ($I_{real}$ or $I_{fake}$) is passed through a series of downsampling blocks ($DownBlocks$), until the feature map is of size $8 \times 8$. Next, these $8 \times 8$ dimensional spatial features are passed through two separate branches: one for extracting features for the adversarial loss and the other for computing image features for multi-modal contrastive loss. For the adversarial branch, the input is passed through a DownBlock, ResBlock, and a fully connected layer to predict the logit to represent if the given image is real or fake. The predicted logit is used as input to an Adversarial Hinge loss (Miyato et al., 2018) $\mathcal{L}_{Adv}^D$ as follows:

$$\begin{aligned}
\mathcal{L}_{Adv}^D = & \mathbb{E}_{x \sim p_{\text{data}}}\left[\max(0, 1 - D(x))\right] \\
& + \mathbb{E}_{\hat{x} \sim p_G}\left[\max(0, 1 + D(\hat{x}))\right]
\end{aligned} \tag{10}$$

Here, $x$ and $\hat{x}$ are real ($I_{real}$) and generated ($I_{fake}$) images.

In the multi-modal contrastive loss branch, the features are passed through a DownBlock, ResBlock, and a linear projection layer to output visual features $f_v$. The multi-modal contrastive loss $\mathcal{L}_{\text{cont}}^D$ takes as input the real image features $f_{v_i}$ and sentence embeddings $S_{D_i}$ and calculates the contrastive loss to increase the mutual information in text and image as follows:

$$\mathcal{L}_{\text{cont}}^D\left(f_{v_i}, S_{D_i}\right) = -\log \frac{\exp\left(Sim\left(f_{v_i}, S_{D_i}\right)\right)}{\sum_{j=1}^N \exp\left(Sim\left(f_{v_i}, S_{D_j}\right)\right)} \tag{11}$$

$\mathcal{L}_{\text{cont}}^D$ is the contrastive loss between real image - text pairs. The final objective function for the Discriminator is defined as:

$$\mathcal{L}_D = \mathcal{L}_{Adv}^D + \lambda_3 \mathcal{L}_{\text{cont}}^D \tag{12}$$

### 3.4 Dual text embedding learning

Embeddings can be viewed as memory representation learned by reducing a loss. Multiple embeddings, each learned by optimising on different losses, will capture various memory representations for the same word. The goal of the dual text embedding setup is to learn generator-side word embeddings $W_G$ (along with its encoder) to capture complex representation of words to aid improve the photo-realism of the generated images and discriminator-side word embeddings $W_D$ (along with its encoder) to capture distinctive features for words to align text-image associativity. To achieve this, we make sure that $W_G$ receives only the gradients from image-generation process whereas $W_D$ receives gradients from the discriminator. Specifically, we formulate the generator-side embedding loss ($\mathcal{L}_{emb}^G$) and the discriminator-side embedding loss ($\mathcal{L}_{emb}^D$) as follows:

$$\mathcal{L}_{emb}^G = \mathcal{L}_G \tag{13}$$

$$\mathcal{L}_{emb}^D = \lambda_3 \mathcal{L}_{\text{cont}}^D \tag{14}$$

Here, $\mathcal{L}_G$ denotes the loss function for the generator $G$, $\mathcal{L}_{\text{cont}}^D$ denotes the multi-modal contrastive loss between real image ($I_{real}$) features and discriminator-side sentence embedding $S_D$ for the given text T.

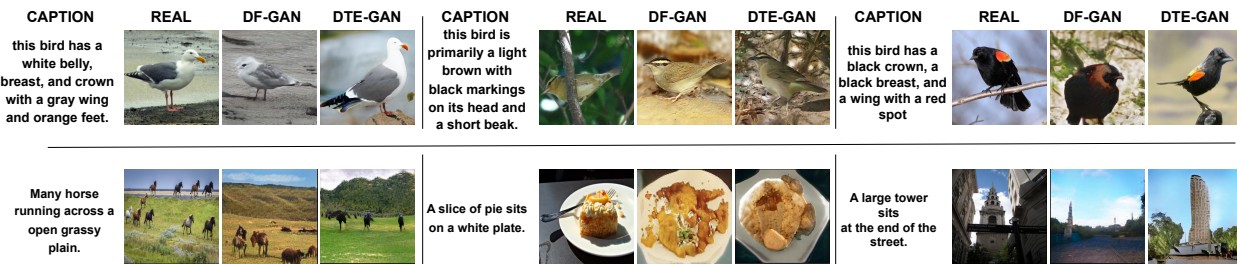

Figure 3: Visual comparision of the images generated by DF-GAN (Tao et al., 2022) and DTE-GAN on CUB(Welinder et al., 2010) and COCO(Lin et al., 2014a) Datasets.

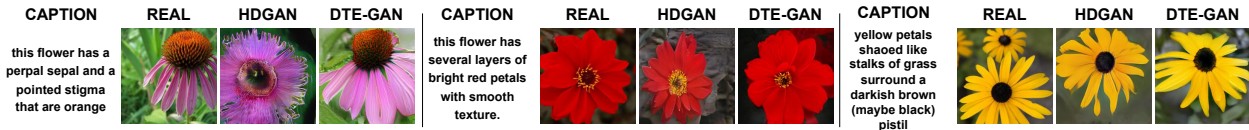

Figure 4: Illustration of the images generated by HDGAN (Zhang et al., 2018) and those of DTE-GAN on Oxford-102 Flower Dataset (Nilsback & Zisserman, 2008).

## 4 Experiments

In this section, datasets and evaluation metrics are introduced for experiments. Futher, proposed DTE is evaluated and compared quantitatively and qualitatively with other methods in the literature. The specific training details and hyper-parameters are mentioned in the supplementary material.

**Datasets:** DTE-GAN is evaluated on three datasets, namely, 1) Caltech-UCSD birds (CUB) (Welinder et al., 2010), 2) Oxford-102 flowers (Nilsback & Zisserman, 2008), and 3) MS COCO (Lin et al., 2014b) datasets. For CUB and Oxford-102 datasets, we have similar setup to StackGAN (Zhang et al., 2017b). Ten captions are provided for each image in both the datasets. The MS-COCO dataset consists of around 80k training and 40k validation images; and for every image, there are 5 captions provided with the dataset.

### 4.1 Visual comparison

The generated images are visually compared between DF-GAN (Tao et al., 2020) and DTE-GAN on CUB and COCO datasets. In Figure 3, it is evident that DF-GAN struggles to depict complete bird shapes. The presented model, employing dual text embeddings, minimises image generation loss, improving shape accuracy and realistic fine-grained features in generated images. Additionally, DTE-GAN outperforms DF-GAN in pose representation, resulting in more natural-looking images. Regarding semantic consistency between images and text, DTE-GAN captures detailed structures and overall coherence compared to those of DF-GAN. Despite not using word embeddings for image region attention, DTE-GAN's ability to learn embeddings for both generation and discrimination allows it to generate images with finer details. DTE-GAN produces images that resemble real images due to its learned embeddings encompassing generation and discrimination aspects. For the COCO dataset, it is observed that DTE-GAN generates similar quality images as DF-GAN with significantly less amount of parameters. In Figure 4, images for the Oxford-102 dataset are generated and compared with those of HDGAN (Zhang et al., 2018). We can observe that our model is able to capture the complex variation of the flowers and generate more realistic images than HDGAN.

### 4.2 Quantitative Evaluation

**Evaluation metrics:** To assess the quality of images generated, the metrics used are: *Inception Score (IS)* (Salimans et al., 2016) and *Fréchet Inception Distance (FID)* (Heusel et al., 2017). For text-to-image

| Method | CUB | | | COCO | | |
|--------|-----|-----|-----|------|-----|-----|
| | IS ↑ | FID ↓ | R% ↑ | FID ↓ | R% ↑ | NoP ↓ |
| StackGAN (Zhang et al., 2017b) | $3.70 \pm .04$ | - | - | - | - | - |
| AttnGAN (Xu et al., 2018) | $4.36 \pm .02$ | 23.98 | 67.82 | 35.49 | 83.82 | $230M$ |
| MirrorGAN (Qiao et al., 2019b) | $4.56 \pm .17$ | 18.32 | 57.67 | 34.71 | 74.53 | - |
| DM-GAN (Zhu et al., 2019) | $4.75 \pm .07$ | 16.09 | 72.32 | 32.64 | 88.56 | $46M$ |
| KT-GAN (Tan et al., 2021) | $4.85 \pm .05$ | 17.32 | - | 30.73 | - | - |
| TIME (Liu et al., 2021) | $4.91 \pm .04$ | 14.30 | 71.57 | 31.14 | - | $120M$ |
| DAE-GAN (Ruan et al., 2021) | $4.42 \pm .04$ | 15.19 | 85.45 | 28.12 | 92.61 | $98M$ |
| CSM-GAN (Tan et al., 2022a) | $4.62 \pm .08$ | 20.18 | - | 33.48 | - | - |
| DR-GAN (Tan et al., 2023a) | $4.90 \pm .05$ | 14.96 | - | 27.80 | - | $73M$ |
| ALR-GAN (Tan et al., 2023b) | $4.96 \pm .04$ | 15.14 | 77.54 | 29.04 | 69.20 | $76M$ |
| DF-GAN (Tao et al., 2022) | $4.86 \pm .04$ | 14.81 | - | 19.32 | - | $19M$ |
| SSA-GAN (Liao et al., 2022) | $5.17 \pm .08$ | 15.61 | 85.4 | 19.37 | 90.6 | $26M$ |
| **DTE-GAN** | $5.12 \pm .04$ | 13.67 | 86.64 | 25.17 | 90.82 | $11M$ |
| **DTE-GAN+MAGP** | $5.09 \pm .02$ | 13.94 | 81.33 | 19.69 | 88.39 | $11M$ |

Table 1: Quantitative comparison between DTE-GAN and other models on CUB (Welinder et al., 2010) and COCO (Lin et al., 2014a) datasets. "-" indicates values are unreported. The best three results are marked with red, green, and blue, respectively. "↑" indicates the higher, the better, while "↓" indicates the lower, the better.

| Method | IS ↑ | FID ↓ |
|--------|------|-------|
| StackGAN (Zhang et al., 2017b) | $3.20 \pm .01$ | 51.89 |
| StackGAN++ (Zhang et al., 2017a) | $3.26 \pm .01$ | 48.68 |
| HDGAN (Zhang et al., 2018) | $3.45 \pm .07$ | 43.17 |
| SegAttnGAN (Gou et al., 2020) | $3.36 \pm .08$ | - |
| SSTIS (Tan et al., 2022b) | $3.37 \pm .05$ | - |
| SS-TiGAN (Tan et al., 2023c) | $3.45 \pm .04$ | 40.54 |
| DualAttn-GAN (Cai et al., 2019) | $4.06 \pm .05$ | 40.31 |
| RAT-GAN (Ye et al., 2024) | 4.09 | - |
| **DTE-GAN** | $4.21 \pm .08$ | 30.07 |
| **DTE-GAN+MAGP** | $4.26 \pm .07$ | 31.13 |

Table 2: Quantitative comparison between DTE-GAN and other models on Oxford-102 Dataset. The best three results are marked with red, green, and blue, respectively. "↑" indicates the higher, the better, while "↓" indicates the lower, the better."-" indicates values are unreported.

alignment, *R-precision* (R%) (Xu et al., 2018) is used. IS calculates the Kullback-Leibler (KL) divergence between a conditional distribution and marginal distribution for class probabilities from Inception-v3 (Szegedy et al., 2016) model. Higher the IS, higher is the quality of images with more diverse classes. FID calculates the Fréchet Distance between two multivariate Gaussians, which are fit to the global features extracted from the Inception-v3 (Szegedy et al., 2016) model on the synthetic and generated images. Lower the FID, higher is the quality of generated images (*i.e.*, closer to real images). R-precision is used to determine the text-to-image alignment, which evaluates whether generated images can be used to retrieve the text.

The performance of proposed model is compared with that of the lightweight GAN approaches (having similar training setups) for the task of text-to-image synthesis on CUB and COCO datasets in Table 1. From Table 1, on the CUB dataset, we observe that DTE-GAN improves *IS* from 4.91 to 5.12, achieves the best *R-precision* of 86.64 and further decreases *FID* from 14.06 to 13.67. We also train our DTE-GAN with the proposed regularisation trick Matching-Aware Gradient Penalty (MAGP) (Tao et al., 2022), which smooths

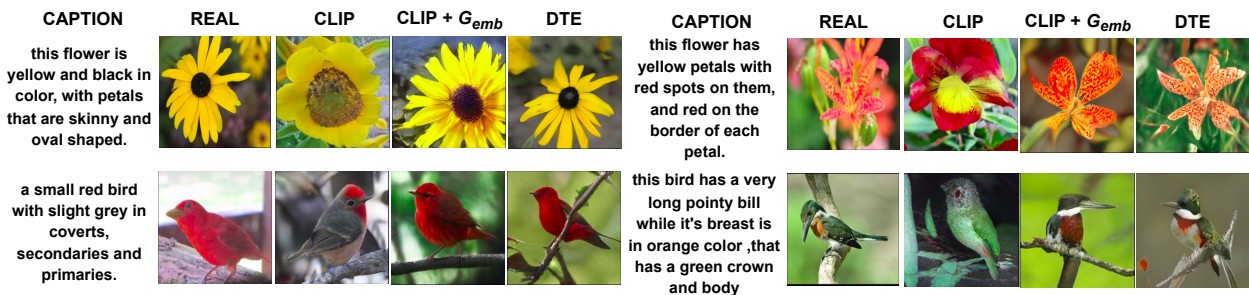

Figure 5: Images generated using CLIP, CLIP + Learnable Generator side embeddings (CLIP + **G_emb**) and DTE on CUB and Oxford-102 datasets.

out discriminator function and allows to generate more realistic images. On the CUB dataset, compared to AttnGAN (Xu et al., 2018) that employs contrastive loss-based embeddings, our model decreases *FID* from 23.98 to 13.67. Such an improvement illustrates the effectiveness of the end-to-end learned dual embeddings over fixed pre-trained embeddings learned on the same data. On MS-COCO (Lin et al., 2014b) dataset (in Table 1), we achieve similar performance of DF-GAN and SSA-GAN with fewer parameters, which proves the efficacy of the proposed DTE. DTE-GAN+MAGP achieves similar performance as that of SSA-GAN (Liao et al., 2022) on COCO dataset as SSA-GAN and DF-GAN (Tao et al., 2022) both incorporating MAGP.

By Capturing both representations in a unified Dual Text Embeddings approach, we achieve superior textual embeddings that improve conditional image generation while reducing the complexity of Text-to-Image synthesis models. As shown in the table, we employed a network with significantly fewer parameters by reducing the width of the layers by half compared to SSA-GAN Liao et al. (2022) and DF-GAN Tao et al. (2020) in their respective UpBlock and DownBlock. Despite this reduction, we achieved comparable results by learning superior text encoding representations that enhance effective Text-to-Image synthesis.

Following previous works (Tao et al., 2022; Zhang & Schomaker, 2020; Liao et al., 2022), we report only *FID* scores, as *IS* scores for the MS-COCO dataset do not reflect the quality of the synthesised images. In comparision to TIME (Liu et al., 2021) which learns embeddings along with the Text-to-Image synthesis model, DTE-GAN achieves significant improvement (0.6 in *FID*, +15% in R-precision) demonstrating the effectiveness of dual text embeddings. In Table 2, on the Oxford-102 dataset, we use *IS* and *FID* scores for evaluation, as R-precision scores are not available in the literature. In this dataset, our model improves *IS* score from 4.06 to 4.21 over the state-of-the-art (DualAttn-GAN (Cai et al., 2019), LeicaGAN (Qiao et al., 2019a)) models and remarkably decreases *FID* from 40.31 to 30.07.

### 4.3   Additional Studies

| Dataset | Embeddings | IS ↑ | FID ↓ | R % ↑ |
|---------|-----------|------|-------|-------|
|         | **CLIP**  | 4.53 | 21.33 | 74.12 |
| **CUB** | **CLIP+G_emb** | 4.51 | 19.36 | 78.35 |
|         | **DTE**   | 5.12 | 13.67 | 86.64 |
|         | **CLIP**  | 3.72 | 38.36 | 71.78 |
| **Oxford** | **CLIP+G_emb** | 3.81 | 35.93 | 73.87 |
|         | **DTE**   | 4.21 | 30.07 | 83.19 |

Table 3: We compare quality of T2I generation of our proposed DTE approach with that of models trained using CLIP (Radford et al., 2021) and CLIP+**G_emb**. The best results are red. "↑" indicates the higher, the better, while "↓" indicates the lower, the better.

### 4.3.1   DTE vs CLIP:

We compare DTE with pre-trainThised CLIP embeddings by training a GAN for text-to-image generation. Additionally, we train another GAN model resembling the DTE setup. This model uses learnable Generator-

side embeddings and pre-trained CLIP embeddings for the discriminator (referred to as CLIP+ $G_{emb}$). Table 3 compares image quality using different CUB and Oxford-102 datasets embeddings. CLIP+**$G_{emb}$** improves over just CLIP. In Figure 5, CLIP images differ from reality, while CLIP+**$G_{emb}$** matches better text and real images due to learned generative embeddings. DTE's combined approach creates images closer to real images.

### 4.3.2 Generalisation ability of DTE:

To assess how well the DTE setup applies to other architectures, we integrate it into AttnGAN (Xu et al., 2018), now called AttnGAN+DTE. In AttnGAN, pre-trained text embeddings (DAMSM embeddings) for training are used. Instead, we have trained it from scratch using DTE. Incorporating DTE into AttnGAN involves modifying its discriminators to include a dual branch (adversarial loss and multi-modal contrastive loss) after reaching $8 \times 8$ feature size. As AttnGAN uses words for alignment in the attention layer, we introduce a word-contrastive loss (Xu et al., 2018; Zhang et al., 2021) in the final discriminator as an additional branch when the feature size is $8 \times 8$, aimed at reducing the semantic gap between words and image features. We combine generator-side and discriminator-side word embeddings to provide word features for the Generator's attention mechanism. The results in Table 4 show that AttnGAN+DTE can train without pre-trained embeddings and even improve the results, proving that DTE works well with other methods.

| Method | IS ↑ | FID ↓ | R% |
|---|---|---|---|
| AttnGAN | $4.36 \pm .02$ | 23.98 | 67.82 |
| AttnGAN+DTE | $4.38 \pm .03$ | 21.45 | 71.39 |
| DTE-GAN | $5.12 \pm .04$ | 13.67 | 86.64 |

Table 4: Impact of DTE approach on AttnGAN (Xu et al., 2018) on CUB Dataset (Welinder et al., 2010). The best results are red. "↑" indicates the higher, the better, while "↓" indicates the lower, the better.

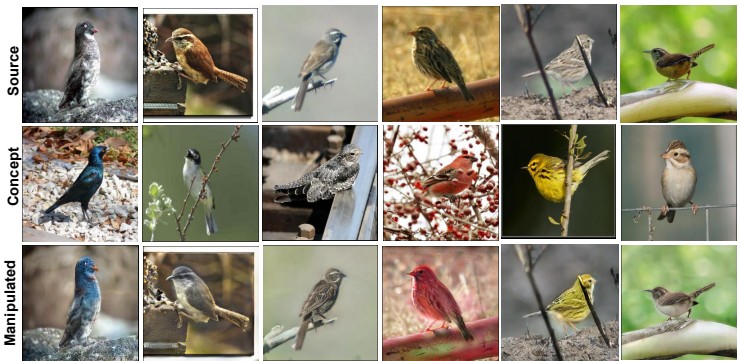

Figure 6: Examples of manipulated images generated by LightWeight GAN (Li et al., 2020b) using DTE-GAN pre-trained embeddings on CUB dataset. Source images are manipulated by the caption of concept images.

### 4.3.3 Application to Text-to-Image manipulation task:

To demonstrate the versatility of the learned dual embeddings, we apply them to text-to-image manipulation tasks. We train dual text embeddings through DTE-GAN on the CUB dataset for text-to-image synthesis. We then utilise the pre-trained word embeddings $(W_G, W_D)$ from this synthesis task for text-to-image manipulation in the Lightweight GAN for Text-to-Image manipulations (Li et al., 2020b) task on the same dataset. It is important to note that these pre-trained dual text embeddings remain fixed during training. In Table 5, the quantitative performance of the model using these pre-trained embeddings is compared with that of other text-to-image manipulation models. The model improves the Inception score from 8.48 (MANIGAN (Li et al., 2020a)) to 8.56 and reduces the *FID* score from 8.02 (Li et al., 2020b) to 7.77 on the CUB dataset. This demonstrates the ability of dual text embeddings to generalise effectively across different tasks. Figure 6 illustrates few visual examples of the text-to-image manipulations.

| Method | IS ↑ | FID ↓ |
|--------|------|-------|
| MANIGAN | 8.48 | 9.75 |
| LWGAN | 8.26 | 8.02 |
| LWGAN w/ DTE-EMB | 8.56 | 7.77 |

Table 5: Quantitative comparison of Inception score and FID for manipulated images on CUB dataset. We use Lightweight GAN (Li et al., 2020b) which we name as LWGAN with the pre-trained embeddings using DTE-GAN (LWGAN w/ DTE-EMB). The best results are red. "↑" indicates the higher, the better, while "↓" indicates the lower, the better.

| Emb. type | Components | | | | CUB | | | Oxford-102 | | |
|-----------|-----------|-----------|-----------|-----------|-----|-----|-----|-----------|-----|-----|
| | $\mathcal{L}_G$ | $\mathcal{L}_{cont}^D$ | $S_D \to G$ | $S_G \to D$ | IS ↑ | FID ↓ | R% ↑ | IS ↑ | FID ↓ | R% ↑ |
| Shared | ✗ | ✓ | - | - | $4.54 \pm .04$ | 15.81 | 85.63 | $3.52 \pm .06$ | 33.57 | 81.73 |
| Shared | ✓ | ✓ | - | - | $4.27 \pm .06$ | 18.38 | 82.73 | $3.32 \pm .05$ | 34.83 | 76.15 |
| Dual | ✓ | ✓ | ✗ | ✗ | $4.73 \pm .05$ | 14.93 | 63.79 | $3.84 \pm .03$ | 32.98 | 54.97 |
| Dual | ✓ | ✓ | ✓ | ✓ | $4.25 \pm .05$ | 18.01 | 69.38 | $3.28 \pm .04$ | 35.41 | 70.48 |
| Dual | ✓ | ✓ | ✓ | ✗ | $5.12 \pm .04$ | 13.67 | 86.64 | $4.21 \pm .08$ | 30.07 | 83.19 |

Table 6: Quantitative comparison of DTE with its variants. Here, $\mathcal{L}_G$ = generator loss, $\mathcal{L}_{cont}^D$ = multi-modal contrastive loss between real image - text pairs, $S_D \to G$ = whether the generator has access to the discriminator-side sentence embedding, $S_G \to D$ = whether the discriminator has access to the generator-side sentence embedding. The best results are red. "↑" indicates the higher, the better, while "↓" indicates the lower, the better.

#### 4.3.4 Importance of dual text embeddings

To verify the effectiveness of the proposed Dual Text Embeddings (DTE) setup, different ways of organising the word embeddings between generator $G$ and discriminator $D$ are evaluated on CUB and Oxford-102 datasets and results shown in Table 6. Specifically, four variants of organising the embeddings are compared, namely: *i*) A shared word embedding layer between $G$ and $D$ that is trained only with a multi-modal contrastive loss $\mathcal{L}_{cont}^D$ between real image-text pairs as it is trained only to capture distinctive features (Table 6, row 1). *ii*) A shared word embedding layer between $G$ and $D$ that is trained using both the generator loss $\mathcal{L}_G$ and real image-text pair contrastive loss $\mathcal{L}_{cont}^D$ to capture distinctive and intricate appearance features in single embeddings (Table 6, row 2). *iii*) A dual embedding setup where $G$ doesn't have access to the discriminator-side sentence embedding $S_D$(Table 6,row 3). *iv*) A dual embedding setup where both $G$ and $D$ have access to the generator-side sentence embeddings $S_G$ and discriminator-side sentence embeddings $S_D$ (Table 6, row 4). The shared embedding model trained only with $\mathcal{L}_{cont}^D$ to capture distinctive features (Table 6, row 1) achieves similar R-precision scores as those of the proposed DTE-GAN, but there is significant drop in *IS* and *FID* scores suggesting that there is drop in the quality of generated images. Next, the shared embedding model trained with both $\mathcal{L}_G$ and $\mathcal{L}_{cont}^D$ performs inferior to the one trained only with $\mathcal{L}_{cont}^D$. It proves that capturing generator-side noisy gradient signals into the same word embeddings degrades the performance. Further, the dual embedding model with independent generator- and discriminator-side embeddings achieves better IS and FID scores compared to those of shared embedding models but has a significant drop in R-precision, suggesting that the images generated are realistic but do not capture text-to-image alignment. Further, allowing the discriminator to have access to the generator-side embeddings significantly drops the performance (Table 6, row 4). As the Discriminator-side embeddings are learned with ground-truth real image-text pairs, it is found beneficial to allow Generator to have access (*i.e.,* a sneak peek) to Discriminator-side embeddings (Table 6, row 5). Discriminator-side embeddings capture distinct features and Generator-side feature captures intricate details to improve photo-realism; providing both the information to Generator allows to generate more realistic and text-aligned images. On the other hand, Generator-side embeddings are learned using noisy gradients from fake images and allowing the discriminator to access them introduces an adverse effect and decreases the performance (Table 6, row 4).

When generator-side embeddings are provided to the discriminator, we observe a significant drop in overall image generation quality, as reported in Table 6. To further evaluate this effect, we conducted an experiment in which generator-side embeddings were supplied to the discriminator after a specified number of training epochs, with the results presented in Table 7. Specifically, generator-side $S_G$ embeddings were combined with $S_D$ using summation from the start of training (epoch = 0), after 100 epochs when the generator began producing images with plausible structure, and after 300 epochs when more realistic images were generated. The low or noisy quality of generated images during the initial stages affects the learning of generator-side embeddings; consequently, providing these embeddings to the discriminator negatively impacts the overall image generation quality.

| Epoch Count | Components | | CUB | | | Oxford-102 | | |
|---|---|---|---|---|---|---|---|---|
| | $S_D \rightarrow G$ | $S_G \rightarrow D$ | IS ↑ | FID ↓ | R% ↑ | IS ↑ | FID ↓ | R% ↑ |
| 0 | ✓ | ✓ | $4.21 \pm .04$ | 18.33 | 69.81 | $3.34 \pm .05$ | 36.18 | 71.13 |
| 100 | ✓ | ✓ | $4.59 \pm .06$ | 16.24 | 74.46 | $3.56 \pm .05$ | 33.27 | 75.68 |
| 300 | ✓ | ✓ | $4.82 \pm .04$ | 14.96 | 78.57 | $3.91 \pm .04$ | 31.38 | 78.92 |

Table 7: Quantitative comparison of of $(S_G \rightarrow D)$ whether the discriminator has access to the generator-side sentence embedding from the epoch count. $S_D \rightarrow G$ represents generator has access to the discriminator-side sentence embedding. The best results are red. "↑" indicates the higher, the better, while "↓" indicates the lower, the better.

## 5 Limitation and Future Scope

This work proposes an approach to learning vision-language models by generating images. Due to computational constraints (our model is trained on a single 1080Ti graphics card with 12 GB memory), we focus our approach and conduct experiments on smaller datasets. For future work, we aim to explore learning vision-language models by generating images using diffusion-based (Ho et al., 2020; Dhariwal & Nichol, 2021) models on large-scale, openly available datasets (Bain et al., 2021; Wang et al., 2022).

## 6 Conclusion

This study introduces a unique approach called Dual Text Embeddings (DTE) for text-to-image synthesis. This method learns word embeddings simultaneously with the main task, enhancing image quality by considering both generation and discrimination aspects separately. DTE-GAN paves a new path in language model design by employing multiple components in embeddings, optimising them independently decoupled. This strategy proves effective for end-to-end learning across various tasks.

In comparision to embeddings trained for distinct representations, our experiments reveal that the DTE setup enhances image quality across three datasets (Oxford-102, Caltect-UCSD, and MS-COCO). It can be seamlessly integrated into existing GAN architectures. Moreover, we showcase the adaptability of learned dual embeddings in different language-based vision tasks, like Text-to-Image manipulations. In our upcoming work, we plan to extend the dual embedding setup to other language-based vision tasks, including image or video captioning and visual question answering.

### 6.1 Broader Impact Statement

The proposed Dual Text Embeddings paradigm introduces a new approach where text encoding schemes capture the perspective of current vision-language models and are also trained to capture generators perspective. This approach enables learning text representations that enhance the image synthesis capabilities of conditional generation models and also provide a new paradigm of learning representation by generating images.

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

## A   DTE-GAN with MA-GP

Recent methods (Tao et al., 2022; Zhang & Schomaker, 2020; Liao et al., 2022) have significantly improved quality of synthesised images on COCO dataset (Lin et al., 2014b). Their improved performance may be associated with the Matching-Aware zero-centered Gradient Penalty (MA-GP) term adopted in these methods. MA-GP is applied to Discriminator with real images and its corresponding text to smooth the loss function that allows the Generator to synthesise more realistic images. Incorporating MA-GP into DTE-GAN by changing the Discriminator to conditional Discriminator (towards one-way output for gradient penalty). Mismatch pairs are not used as DTE-GAN has a multi-modal contrastive branch for text-image alignment. For conditional prediction, following similar setup of DF-GAN (Tao et al., 2022) of replicating sentence features and concatenating with image features to predict logit values for adversarial loss. The modified adversarial loss function of Discriminator for conditional loss with MA-GP is:

$$
\begin{aligned}
L_D^{Adv} = &- \mathbb{E}_{x \sim \mathbb{P}_{data}}[\max(0, 1 - D(x, S_D))] \\
&+ \mathbb{E}_{\hat{x} \sim \mathbb{P}_G}[\max(0, 1 + D(\hat{x}, S_D))] \\
&+ k\mathbb{E}_{x \sim \mathbb{P}_{data}}\left[(\|\nabla_x D(x, S_D)\| + \|\nabla_{S_D} D(x, S_D)\|)^p\right]
\end{aligned}
\tag{15}
$$

Here, $k$ and $p$ are hyper-parameters (we use the same hyper-parameter values from DF-GAN (Tao et al., 2022)). For training discriminator-side word embeddings and their sentence encoder from real image-text pairs, we do not update the weights using the gradient of fake conditional prediction from the adversarial loss. Conditional Adversarial loss for Generator is:

$$
L_G^{Adv} = - \mathbb{E}_{\hat{x} \sim \mathbb{P}_G}[D(\hat{x}, S_D)]
\tag{16}
$$

The final objective function for the Generator and Discriminator is defined as:

$$
\mathcal{L}_G = \mathcal{L}_{Adv}^G + \lambda_1 \mathcal{L}_{CA} + \lambda_2 \mathcal{L}_{cont}^G
\tag{17}
$$

$$
\mathcal{L}_D = \mathcal{L}_{GAN}^D + \lambda_3 \mathcal{L}_{cont}^D
\tag{18}
$$

## B  Text Encoding Scheme

We used a single-stage Bi-LSTM Schuster & Paliwal (1997) for text encoding, following the popular DAMSM Xu et al. (2018) embeddings commonly employed in lightweight GAN models Xu et al. (2018); Zhu et al. (2019); Liao et al. (2022); Tao et al. (2020). The DAMSM embeddings are trained to learn discriminative features by distinguishing between instances, ensuring a fair comparison focused on design principles rather than simply increasing the number of parameters. Additionally, we conducted an ablation study by replacing the Bi-LSTM (Schuster & Paliwal, 1997) with a 4-layer Transformer encoder Vaswani et al. (2017) in both the generator and discriminator text encoders, and we reported the results in Table 8.

| Dataset | Encodings | IS ↑ | FID ↓ | R % ↑ |
|---------|-----------|------|-------|-------|
| CUB | **Bi-LSTM** | 5.12 | 13.67 | 86.64 |
|  | **Transformer Encoder** | 5.19 | 13.12 | 87.9 |
| Oxford | **Bi-LSTM** | 4.21 | 30.07 | 83.19 |
|  | **Transformer Encoder** | 4.27 | 29.61 | 83.94 |

Table 8: We compare quality of T2I generation using Bi-LSTM and 4 layer Transformer Encoder text encoding scheme and report the results on CUB and Oxford-102 dataset.

## C  Details of the Proposed Architecture

In this section, we elaborate internal architecture details of the DTE-GAN. Proposed model is implemented using Pytorch (Paszke et al., 2019) framework. DTE-GAN architecture consists of a dual text embedding setup (Section C.0.1), a single-stage Generator (Section C.0.2) and a Discriminator (Section C.0.3).

### C.0.1  Dual Text Embeddings

In the Dual Text Embeddings setup, bi-Directional LSTM (Schuster & Paliwal, 1997) are used as text encoder both generator-side and discriminator-side. For each direction in the LSTM, hidden layer size is set as 128. The size of word embeddings $W_D$ and $W_G$ is set to 256. The sentence embeddings $S_G$, $S_D$ are encoded from the output of last hidden state of respective text encoders. For both the text encoders, sentence embedding size is set to 256.

### C.0.2  Generator

Single-stage generator $G$ is used to generate $256 \times 256$ resolution images with base channel dimension of 64. Details of $G$'s architecture are shown in Table 9. The generator $G$ takes noise $z$ along with generator-side sentence embeddings $S_G$ and the discriminator-side sentence embedding $S_D$ and passes them through a set of linear layers followed by a set of upsampling blocks (UpBlocks). UpBlock at each stage is utilised for up sampling spatial features as shown in Figure 7. $S_G$ and $S_D$ are also used to calculate modulation parameters for Conditional Batch Normalisation (Yin et al., 2019). Feature are passed through a self modulation convolution and a $1 \times 1$ convolution resulting in generation of final image of dimension $3 \times 256 \times 256$

### C.0.3  Discriminator

Discriminator $D$ is utilise to provide adversarial loss and also act as a feature extractor for multi-modal contrastive loss (as shown in Table 10). Unlike multiple / multi-stage discriminator setup, presented model with a single discriminator, is easy to train and not having a cumbersome training procedure. The discriminator $D$ takes image of dimension $3 \times 256 \times 256$ and passes it through a set of down sampling blocks (DownBlocks- as shown in Figure 8) followed by two branches, one for the adversarial loss and the other multi-modal contrastive loss.

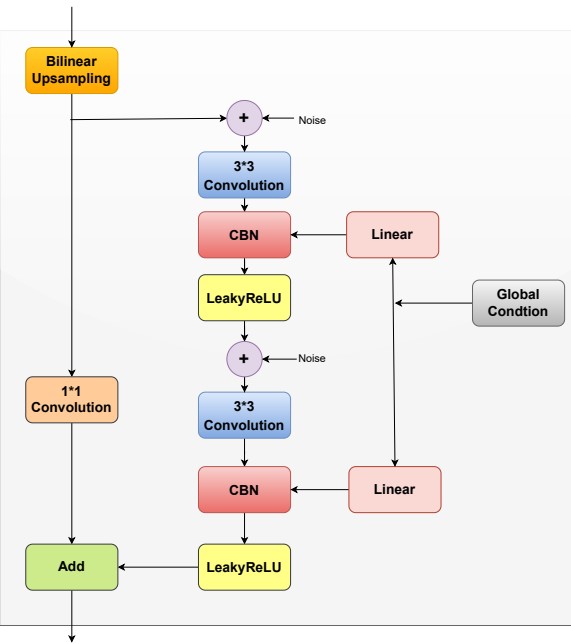

Figure 7: UpBlock used in Generator of DTE-GAN.

| $z\epsilon\mathbb{R}^{100} \sim \mathcal{N}(0, I)$ , $S_G \epsilon \mathbb{R}^{256}$, $S_D \epsilon \mathbb{R}^{256}$, $W_G \epsilon \mathbb{R}^{256}$, $W_D \epsilon \mathbb{R}^{256}$ |
|:---:|
| Linear(512) $\longrightarrow$ 512 |
| Conditional Augmentation(512) $\longrightarrow$ 200 |
| Linear(200+100) $\longrightarrow$ $(8 * ch) \times 4 \times 4$ |
| UpBlock $\longrightarrow$ $(8 * ch) \times 8 \times 8$ |
| UpBlock $\longrightarrow$ $(8 * ch) \times 16 \times 16$ |
| UpBlock $\longrightarrow$ $(4 * ch) \times 32 \times 32$ |
| UpBlock $\longrightarrow$ $(2 * ch) \times 64 \times 64$ |
| UpBlock $\longrightarrow$ $(2 * ch) \times 128 \times 128$ |
| UpBlock $\longrightarrow$ $ch \times 256 \times 256$ |
| Self Modulation Convolution $\longrightarrow$ $ch \times 256 \times 256$ |
| $1 \times 1$ Convolution $\longrightarrow$ $3 \times 256 \times 256$ |

Table 9: Generator architecture of DTE-GAN. Base channel dimension $ch = 64$.

## C.1 Implementation Details

Implementation of the models is done using the PyTorch framework (Paszke et al., 2019) and optimise the network using Adam optimiser (Kingma & Ba, 2015) with the following hyper parameters: $\beta_1 = 0.5$, $\beta_2 = 0.999$, batch size = 24, learning rate = 0.0002, $\lambda_1 = 1$, $\lambda_2 = 1$ and $\lambda_3 = 1$. Spectral Normalisation (Miyato et al., 2018) is used for all convolutions and fully connected layers in generator and discriminator. The model is trained for 600 epochs on CUB and Oxford-102 datasets (takes ~4 days in 2 NVIDIA 1080Ti GPUs) and 120 epochs for COCO dataset (takes ~7 days in 2 NVIDIA 1080Ti GPUs). During inference,

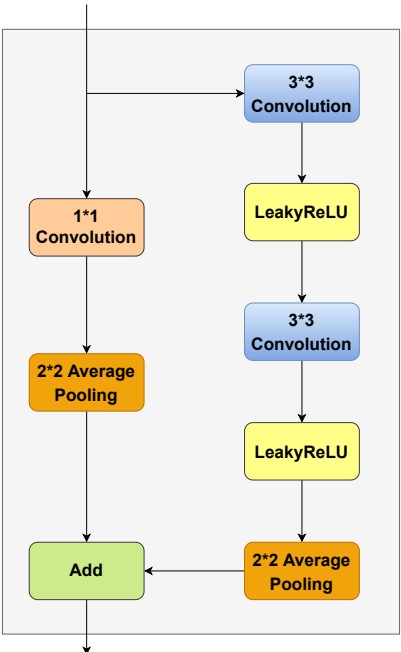

Figure 8: DownBlock used in Discriminator of DTE-GAN.

| RGB images $3 \times 256 \times 256$, $S_D \ \epsilon \ \mathbb{R}^{256}$, $W_D \ \epsilon \ \mathbb{R}^{256}$ | |
|:---:|:---:|
| DownBlock $\longrightarrow ch \times 128 \times 128$ | |
| DownBlock $\longrightarrow (2 * ch) \times 64 \times 64$ | |
| DownBlock $\longrightarrow (4 * ch) \times 32 \times 32$ | |
| DownBlock $\longrightarrow (4 * ch) \times 16 \times 16$ | |
| DownBlock $\longrightarrow (4 * ch) \times 8 \times 8$ | |
| DownBlock $\longrightarrow (8 * ch) \times 4 \times 4$ | DownBlock $\longrightarrow (8 * ch) \times 4 \times 4$ |
| ResBlock $\longrightarrow (8 * ch) \times 4 \times 4$ | ResBlock $\longrightarrow (8 * ch) \times 4 \times 4$ |
| Linear$((8 * ch) \times 4 \times 4) \longrightarrow 1$ | Linear$((8 * ch) \times 4 \times 4) \longrightarrow 256$ |
| Adversarial Loss | Multi-Modal Contrastive Loss |

Table 10: Discriminator architecture of DTE-GAN. Base channel dimension $ch = 64$.

we report results with exponential moving average weights, with a decay rate of 0.999. For R-precision, we obtain text features from D-side Bi-LSTM sentence encoder and image features from discriminator network.

