# OpenReview forum: "Learning Dual Text Embeddings by Synthesising Images Conditioned on Text"
_TMLR — Rejected by TMLR_

### Review · Reviewer_tAQL · 2024-08-29

**Summary Of Contributions:**

this paper explores a new direction to learn text embeddings in an end-to-end manner from text-to-image synthesis task that considers the complementary nature of generation and discrimination process. Specifically, a novel text-embedding learning scheme called "Dual Text Embedding" (DTE) is presented, in which one part of the embeddings is optimised to enhance the photo-realism of the generated images, and the other part seeks to capture text-to-image alignment. Through a comprehensive set of experiments on three text-to-image benchmark datasets (Oxford-102, Caltech-UCSD, and MS-COCO), models with dual text embeddings perform favourably in comparison with
embeddings trained only to learn distinctive features.

**Audience:**

Yes

**Broader Impact Concerns:**

NAN

**Claims And Evidence:**

Yes

**Requested Changes:**

If the author can provide a reasonable explanation for the above-mentioned deficiencies and appropriately supplement them in the submitted manuscript, then the paper may be accepted.

**Strengths And Weaknesses:**

++The paper  is well organized.

++A large number of experiments have verified the effectiveness of the algorithm.

Nonetheless, I still have some doubts that require the author to address.

-- In Table 3, we are curious whether CLIP is trained with GAN based on T2I data. If not, is such a comparison fair?

--What is the resolution of the generated image. If the resolution of generated images becomes higher, can we guarantee their quality? What will happen to the model's volume?

--What text encoding model is used? How does the performance of different text encoding models compare to that of the model?

--Please explain the principle of DTE design? Why is this design? This part of the explanation is suggested to be added in Introduction

---

> ### Author Response · Authors · 2024-10-04
> **Response to Reviewer tAQL (Part-1)**
>
> We thank the reviewer tAQL for their valuable time and constructive feedback. We have carefully incorporated the suggestions into the revised manuscript, and our responses to each concern are detailed below.
>
> Q1. In Table 3, we are curious whether CLIP is trained with GAN based on T2I data. If not, is such a comparison fair?
>
> A1. We use a pre-trained CLIP [1] model to extract sentence embeddings for captions, following the widely adopted practice of leveraging pre-trained vision-language models for caption representation in the Text-to-Image space [2,3,4,5,6,10]. The comparison emphasises learning generator-side embeddings, explicitly trained by generating images to improve image quality. The table demonstrates that combining CLIP embeddings with embedding learned by generating images leads to enhanced performance.
>
> Q2. What is the resolution of the generated image. If the resolution of generated images becomes higher, can we guarantee their quality? What will happen to the model's volume?
>
> A2. We adopted earlier GAN-based [2,4,5] approaches to generate images at a resolution of 256×256. Our network design incorporates upsampling blocks in the generator and downsampling blocks in the discriminator to manage image resolution. Using a single generator, some methods have achieved higher resolutions, such as 512×512 [10,12]. However, increasing the resolution significantly raises GPU memory requirements, limiting us to training on a single 1080 Ti graphics card. Incorporating these blocks in the generator and discriminator allows for higher-resolution image generation.
>
> Q3. What text encoding model is used? How does the performance of different text encoding models compare to that of the model?
>
> We used a single-stage Bi-LSTM [7] for text encoding, following the popular DAMSM [2] embeddings commonly employed in lightweight GAN models  [2,4,5]. The DAMSM embeddings are trained to learn discriminative features by distinguishing between instances, ensuring a fair comparison focused on design principles rather than simply increasing the number of parameters. Additionally, we conducted an ablation study by replacing the Bi-LSTM [7] with a 4-layer Transformer encoder [8] in both the generator and discriminator text encoders, and we reported the results in Table 8. We observed a marginal performance improvement, as shown in the Appendix section B of the revised paper.

---

> > ### Author Response · Authors · 2024-10-04
> > **Response to Reviewer tAQL (Part-2)**
> >
> > Q4. Please explain the principle of DTE design? Why is this design? This part of the explanation is suggested to be added in Introduction
> >
> > Recent state-of-the-art methods [2,3,4,5,6,9,10,12] are predominantly trained using pre-trained vision-language models [1,2]. These vision-language models project images and text into a shared embedding space, where matching pairs exhibit high similarity and dissimilar pairs show low similarity. This training relies on contrastive learning to distinguish between pairs by learning discriminative features as shown in Figure 1.a, resulting in superior text-image alignment. Such models are commonly used for encoding captions in Text-to-Image synthesis. This scheme of learning embeddings comes with a limitation that these learned text embeddings are unaware of the different perspectives of generation and discrimination process, i.e., the generative process necessitates learning complex appearance variations needed for creating realistic images. Integrating generative and discriminator perspectives leads to a superior text encoding scheme, enhancing Text-to-Image synthesis.
> >
> > In this work, we explore a new direction to capture both such perspectives in an unified end-to-end training as shown in Figure 1.c for improving compatibility between image and text modalities. In this regard, a novel Dual Text Embedding GAN (DTE-GAN) setup is proposed to learn text embeddings in an end-to-end manner that takes into account the different perspectives of the generation and discrimination process.
> >
> > By capturing both representations in a unified Dual Text Embeddings approach, we achieve superior textual embeddings that improve conditional image generation while reducing the complexity of Text-to-Image synthesis models. As shown in the table, we employed a network with significantly fewer parameters by reducing the width of the layers by half compared to SSA-GAN [4] and DF-GAN [5] in their respective UpBlock and DownBlock. Despite this reduction, we achieved comparable results by learning superior text encoding representations that enhance effective Text-to-Image synthesis. Further, ablation study regarding the design choice of the network is performed in the section "Importance of Dual Text Embeddings".
> >
> > ***
> > References
> >
> > [1] Radford, Alec, et al. "Learning transferable visual models from natural language supervision." International conference on machine learning. PMLR, 2021.
> >
> > [2] Xu, Tao, et al. "Attngan: Fine-grained text to image generation with attentional generative adversarial networks." Proceedings of the IEEE conference on computer vision and pattern recognition. 2018.
> >
> > [3] Kang, Minguk, et al. "Scaling up gans for text-to-image synthesis." Proceedings of the IEEE/CVF Conference on Computer Vision and Pattern Recognition. 2023.
> >
> > [4] Liao, Wentong, et al. "Text to image generation with semantic-spatial aware gan." Proceedings of the IEEE/CVF conference on computer vision and pattern recognition. 2022.
> >
> > [5] Tao, Ming, et al. "Df-gan: A simple and effective baseline for text-to-image synthesis." Proceedings of the IEEE/CVF conference on computer vision and pattern recognition. 2022.
> >
> > [6] Rombach, Robin, et al. "High-resolution image synthesis with latent diffusion models." Proceedings of the IEEE/CVF conference on computer vision and pattern recognition. 2022.
> >
> > [7] Sak, H. "Long short-term memory based recur rent neural network architectures for large vocabulary speech recognition." arXiv preprint arXiv:1402.1128 (2014).
> >
> > [8] Vaswani, A. "Attention is all you need." Advances in Neural Information Processing Systems (2017).
> >
> > [9] Ramesh, Aditya, et al. "Zero-shot text-to-image generation." International conference on machine learning. Pmlr, 2021.
> >
> > [10] Sauer, Axel, et al. "Stylegan-t: Unlocking the power of gans for fast large-scale text-to-image synthesis." International conference on machine learning. PMLR, 2023.
> >
> > [11] Ramesh, Aditya, et al. "Hierarchical text-conditional image generation with clip latents." arXiv preprint arXiv:2204.06125 1.2 (2022): 3.
> >
> > [12] Zhang, Zizhao, Yuanpu Xie, and Lin Yang. "Photographic text-to-image synthesis with a hierarchically-nested adversarial network." Proceedings of the IEEE conference on computer vision and pattern recognition. 2018.

---

### Review · Reviewer_AyLb · 2024-09-02

**Summary Of Contributions:**

The paper proposes a dual-embedding setup, DTE, for text-to-image GANs, where a generator-side text encoder and a discriminator-side text encoder are learned with different contrastive losses. Experiments show that the dual-embedding setup improves both the image quality and the text-image alignment.

**Audience:**

Yes

**Claims And Evidence:**

Yes

**Requested Changes:**

- References are not cited in the correct format. They should be enclosed within parentheses.
- Typo: "hese" should be "these" in 6th line in page 2.
- Citation missing in 4th line in In Section 5.

**Strengths And Weaknesses:**

Strengths:
- The paper demonstrates various experiments to support the efficacy of the proposed method.

Weaknesses:
- Writing can be improved.
- It is claimed that DTE captures the complementary nature of generation and discrimination. However, no analysis about this complementary nature is provided, nor are there experimental results demonstrating that DTE's dual embeddings work complementarily.
- The framework seems to be compatible only with GAN models.

---

> ### Author Response · Authors · 2024-10-04
> **Response to Reviewer AyLB**
>
> We thank the reviewer tAQL for their valuable time and constructive feedback. We have carefully incorporated the suggestions into the revised manuscript, and our responses to each concern are detailed below.
>
> Q1. It is claimed that DTE captures the complementary nature of generation and discrimination. However, no analysis about this complementary nature is provided, nor are there experimental results demonstrating that DTE's dual embeddings work complementarily.
>
> A1. We focus on capturing different representations of words from the perspectives of discrimination and generation. In the CLIP vs. DTE experiment, we utilise CLIP embeddings, which are trained to distinguish between instances by learning discriminative features. While these embeddings provided to generators effectively create images capturing the overall semantics of the text, while these images might reflect the actual image. To bridge this gap, embeddings learned through image generation capture intricate aspects that facilitate creating images closer to real ones.
>
> We have updated the 'complementary nature of generations' with 'different perspective of generations,' as 'complementary' implies all other aspects that discriminator-side learnings may miss. We thank the reviewers for allowing us to reflect on this.
>
> Q2. The framework seems to be compatible only with GAN models.
>
> A2. We have proposed an approach highlighting the importance of having both perspectives of representation for generating images. Our unified method learns both embeddings in an end-to-end manner. Furthermore, we demonstrate that this approach can be integrated with pre-trained vision-language models, specifically CLIP + $G_{emb}$, where only the generator-side embeddings are learned. This method of focusing solely on generator-side embeddings and it can be seamlessly incorporated into current diffusion-based Text-to-Image models [1] within the existing training frameworks.
>
> ***
> References
>
> [1] Rombach, Robin, et al. "High-resolution image synthesis with latent diffusion models." Proceedings of the IEEE/CVF conference on computer vision and pattern recognition. 2022.

---

### Review · Reviewer_YgyL · 2024-09-20

**Summary Of Contributions:**

The paper proposes a a dual text embedding (DTE) method to separately optimize embeddings based on their function. The proposed method demonstrates good performance compared with CLIP,  AttnGAN, as well as on text-to-image manipulation task.

**Audience:**

Yes

**Claims And Evidence:**

No

**Requested Changes:**

Please see above weaknesses and do necessary midifications.

**Strengths And Weaknesses:**

Strengths:
1. The idea of dual text embedding is interesting and also general. I feel it could be adapted to other multi-modal tasks as well.

2. The proposed method illustrated good performance compared with some strong baseline models, such as CLIP.

Weaknesses:
1. Some parts of this paper are not quite clear to me. For example, the author emphasized "The learned text embeddings by previous methods  are unaware of the complementary nature of generation, while their method could alleviate this issue." But it's unclear to me that what the specific meaning of the complementary nature is, and how the proposed method could solve this issue.

2. Some hypotheses have not been verified. For example, in Introduction, the paper mentioned "naive approaches of independent or fully shared text embeddings between the generator and discriminator lead to sub-par results. We hypothesise that one of the prominent reasons for such sub-par results is the noisy gradient during GAN training." Then the authors built their method based on this hypothesis, but they didn't do some analysis to verify if this hypothesis is true.

3. Some claims seems not quite accurate to me, such as in Section 3.2, a single-stage generator is employed that can generate an image at any resolution owing to its simpler design and easier training procedure. I doubt if this claim is true.

4. The paper lacks ablation study.

5. In Table1, the method shows mixed performance, why is that?

6. In Table2, the latest baseline is a 2019 paper. The authors might need to compare their method with some recent SOTA models.

---

> ### Author Response · Authors · 2024-10-04
> **Response to Reviewer YgyL (Part-1)**
>
> We are thankful to the reviewer for their insightful suggestions, which have significantly improved our manuscript. We have implemented the changes accordingly, and our responses to the specific comments are provided below.
>
> Q1. Some parts of this paper are not quite clear to me. For example, the author emphasised "The learned text embeddings by previous methods are unaware of the complementary nature of generation, while their method could alleviate this issue." But it's unclear to me that what the specific meaning of the complementary nature is, and how the proposed method could solve this issue.
>
> A1. We focus on capturing different representations of words from the perspectives of discrimination and generation. In the CLIP vs. DTE experiment, we utilise CLIP embeddings, which are trained to distinguish between instances by learning discriminative features. While these embeddings provided to generators effectively create images capturing the overall semantics of the text, while these images might reflect the actual image. To bridge this gap, embeddings learned through image generation capture intricate aspects that facilitate creating images closer to real ones.
>
> We have updated the 'complementary nature of generations' with 'different perspective of generations,' as 'complementary' implies all other aspects that discriminator-side learnings may miss. We thank the reviewers for allowing us to reflect on this.
>
>
> Q2. Some hypotheses have not been verified. For example, in Introduction, the paper mentioned "naive approaches of independent or fully shared text embeddings between the generator and discriminator lead to sub-par results. We hypothesise that one of the prominent reasons for such sub-par results is the noisy gradient during GAN training." Then the authors built their method based on this hypothesis, but they didn't do some analysis to verify if this hypothesis is true.
>
> A2. When generator-side embeddings are provided to the discriminator, we observe a significant drop in overall image generation quality, as reported in Table 6. To further evaluate this effect, we conducted an experiment in which generator-side embeddings were supplied to the discriminator after a specified number of training epochs, with the results presented in Table 7. Specifically, generator-side $S_G$ embeddings were combined with $S_D$ using summation from the start of training (epoch = 0), after 100 epochs when the generator began producing images with plausible structure, and after 300 epochs when more realistic images were generated. The low or noisy quality of generated images during the initial stages affects the learning of generator-side embeddings; consequently, providing these embeddings to the discriminator negatively impacts the overall image generation quality.
>
> Q3. Some claims seems not quite accurate to me, such as in Section 3.2, a single-stage generator is employed that can generate an image at any resolution owing to its simpler design and easier training procedure. I doubt if this claim is true.
>
> A3. We adopted earlier GAN-based [2,3,4] approaches to generate images at a resolution of 256×256. Our network design incorporates upsampling blocks in the generator and downsampling blocks in the discriminator to manage image resolution. Using a single generator, some methods have achieved higher resolutions, such as 512×512 [5,6].

---

> > ### Author Response · Authors · 2024-10-04
> > **Response to Reviewer YgyL (Part 2)**
> >
> > Q4. In Table1, the method shows mixed performance, why is that?
> >
> > A4. As shown in the table, we employed a network with significantly fewer parameters by reducing the width of the layers by half compared to SSA-GAN [3] and DF-GAN [4] in their respective UpBlock and DownBlock. Despite this reduction, we achieved comparable results by learning superior text encoding representations that enhance effective Text-to-Image synthesis.
> >
> >
> > Q5. In Table2, the latest baseline is a 2019 paper. The authors might need to compare their method with some recent SOTA models
> >
> > A5. We have included a comparison with recent methods in Table 2 in the revised submission.
> >
> > ***
> > References
> >
> >
> > [1] Radford, Alec, et al. "Learning transferable visual models from natural language supervision." International conference on machine learning. PMLR, 2021.
> >
> > [2] Xu, Tao, et al. "Attngan: Fine-grained text to image generation with attentional generative adversarial networks." Proceedings of the IEEE conference on computer vision and pattern recognition. 2018.
> >
> > [3] Liao, Wentong, et al. "Text to image generation with semantic-spatial aware gan." Proceedings of the IEEE/CVF conference on computer vision and pattern recognition. 2022.
> >
> > [4] Tao, Ming, et al. "Df-gan: A simple and effective baseline for text-to-image synthesis." Proceedings of the IEEE/CVF conference on computer vision and pattern recognition. 2022.
> >
> > [5] Sauer, Axel, et al. "Stylegan-t: Unlocking the power of gans for fast large-scale text-to-image synthesis." International conference on machine learning. PMLR, 2023.
> >
> > [6] Zhang, Zizhao, Yuanpu Xie, and Lin Yang. "Photographic text-to-image synthesis with a hierarchically-nested adversarial network." Proceedings of the IEEE conference on computer vision and pattern recognition. 2018.

---

### Decision · Action_Editor_gzZw · 2024-11-21

**Recommendation:** Reject

**Comment:**

As discussed above, while the reviewers acknowledged the good results reported in the paper, they were unconvinced by the discussion of complementarity in the paper, even after the authors' rebuttal, and felt that this aspect of the work was not supported by clear evidence. As this corresponds to a key contribution of this work, it needs to be thoroughly validated, and therefore the paper cannot be accepted.

**Audience:**

The topic studied in this paper is definitely of interest to the TMLR audience. However, the lack of evidence for one of the main claims will make it less attractive.

**Claims And Evidence:**

Although the reviewers acknowledge the good results reported in the paper, they argue that the claims are not all supported by clear evidence. In particular, in their reviews, all the reviewers questioned the motivation behind this work, and specifically the complementarity of the learned embeddings. The authors revised their statement during the discussion period, replacing 'complementary nature of generations' with 'different perspective of generations'. Unfortunately, this did not convince the reviewers, who, in their final recommendations argued that:
- the authors "still didn't specify what these 'different perspective' are, nor provide analysis and justification about it." (Reviewer AyLb)
- the reviewer has "some doubts about the claims in this paper" and thinks "this paper lacks some necessary experiments."(Reviewer YgyL)